# Condition Monitoring of the Dampers in the Railway Vehicle Suspension Based on the Vibrations Response Analysis of the Bogie

**DOI:** 10.3390/s22093290

**Published:** 2022-04-25

**Authors:** Mădălina Dumitriu

**Affiliations:** Department of Railway Vehicles, University Politehnica of Bucharest, 060042 București, Romania; madalina.dumitriu@upb.ro

**Keywords:** railway vehicle, fault detection, primary suspension, damper, vertical vibration

## Abstract

This paper investigates the possibility of developing a new method for fault detection of a damper in the primary suspension of the railway vehicle, based on the analysis of the vertical vibration’s response of the bogie. To this purpose, experimental data are used, along with results from numerical simulations regarding the Root Mean Square (RMS) accelerations measured/simulated in four reference bogie points—two points on the chassis, against the suspension, and two points located against the axle boxes. The experimental data are utilized to define the normal area of operating and the damper failure area in the bogie primary suspension, as well as a basis for validating the results of numerical simulations. The numerical simulations are developed on the basis of two original models of the vehicle–track system, rigid-flexible coupled type, which take into account the elasticity of the vehicle carbody and the elasticity of the wheel-rail contact: a reference model with 15 degrees of freedom, for simulating the bogie response to vertical vibrations for the normal operating of the primary suspension dampers, and an extended model with 20 degrees of freedom, for simulating the bogie vibration response to the failure damper of a primary suspension. The presented results show that there are clear premises on the possibilities of developing a fault detection method of any of the four dampers of the primary suspension corresponding to a vehicle bogie, based on the RMS accelerations measured only in two reference points of the bogie.

## 1. Introduction

To respond to the increasing mobility standards at all times, the railway operators are constantly concerned about providing availability and punctuality of the transportation services being supplied. That implies avoiding the unplanned interruptions and minimization of the railway vehicles’ immobilization with the purpose of conducting specific maintenance activities.

The maintenance plans for the railway vehicles often rely on techniques to involve scheduled preventive maintenance—calendar-based maintenance, usually mixed with infrequent and simple visual inspections. The maintenance activities of the railway vehicles based on the calendar are carried out periodically, the maintenance intervals being defined according to the time or kilometers covered. The selection of these intervals is based on theoretical considerations and experience, which can lead to too short maintenance intervals. This maintenance method is not always able to identify and detect emergent faults, thus leading to faults between the planned maintenance times. In addition, this method brings about the replacement of the fully operational components, as they have reached their time or distance limit [1].

A solution to minimize the vehicle immobilization time is condition-based maintenance. Unlike calendar-based maintenance, where the vehicle component elements are replaced after certain pre-defined intervals, the condition-based maintenance system requires that the elements be changed according to their real condition [2]. This requires the detection of defects that occur during component operation based on changes in their dynamic properties. The technique that examines the dynamic behavior of a vehicle during operation is known as condition monitoring [3,4,5].

Condition monitoring can be regarded as a well-established part of the fault detection and isolation/identification area. This technique is mainly applied to the systems deteriorating over time (e.g., mechanical systems that have well-defined deterioration mechanisms (wear, fatigue, etc.) and implies that the fault detection and isolation be conducted prior to deterioration. This is actually the key element of condition-based maintenance) [6,7,8].

Condition monitoring provides a number of benefits for railway operators. By detecting, in the early stages, some malfunctions in the operation of the vehicle components, the deterioration of its performance is further prevented. The timely repairs or replacement of the faulty components increase the reliability and the availability of the railway vehicle [9]. Last, but not least, the costs associated with the railway vehicles maintenance can be significantly reduced through the introduction of condition-based maintenance instead of calendar-based maintenance [10,11,12,13].

Thanks to its significant benefits, the condition-based monitoring issue of railway vehicles has piqued scientific and industrial interest in recent years. In this broad field of research, the issue of fault detection and identification in the suspension system of the railway vehicle holds an important place. On the one hand, the reliability of the suspension system of the railway vehicle is essential for meeting the requirements imposed to ensure the ride quality and ride comfort and, on the other hand, for complying with the safety requirements [14]. The failure of the suspension components generally prompts a decrease or increase in the nominal values of the damping parameters or the stiffness coefficients. Any deviation from the nominal characteristics of suspension affects the dynamic behavior of the entire vehicle or, in extreme cases, can endanger the passengers ‘safety [5].

The issue of detecting and identifying the defects in the suspension system of the railway vehicle is found in the review literature as a two approaches-model-based approach and a data-driven approach [4,15,16]. 

The model-based approaches are used when there is no possibility to measure the monitored parameters in real-time, but the relationship between input and output data is available [4]. This approach usually requires the development of a complex vehicle model in order to determine the relationship between the faulty states and the dynamic response of the vehicle. The measured data are then introduced in this model to foretell the dynamic response of the vehicle [15]. The results derived from the model are later compared with the measured data, and the defects are identified based on the residue between the measured data and the prediction data. As a rule, the model-based approaches are developed based on the following methods: observer/Kalman filter based method [7,17,18,19,20,21,22,23,24,25,26,27,28]; interacting multiple models based method [29,30,31]; Rao–Blackwellised particle filter-based method [32,33,34]; and recursive least squares based method [35,36]. 

In case only the output data are available, data-based methods are used. The measured data are examined as a response to a form of disturbance in the time and/or frequency domain to identify characteristics associated with certain defects. The band-pass filter, the spectral analysis, the maximum-entropy estimation, and the wavelet analysis are applied as processing methods for the measured data [4,37]. The data-based approaches to detect and identify the defects in the suspension system of the railway vehicle are based on statistical models [38,39,40,41,42,43,44,45,46,47,48], traditional machine learning [49,50,51,52], hybrid model [16,53], and deep learning [54].

The paper investigates the possibility of developing a new method to detect damper faults of the primary suspension of the railway vehicle, which is based on the analysis of the response to the bogie vertical vibrations, expressed as RMS accelerations in four reference points of the bogie. The four reference bogie points are located as such—two points are on the chassis against the supporting points on the primary suspension, and two points are against the axle boxes. The study is based on both experimental data and on the results from numerical simulations developed on the basis of two original models of the vehicle–track system. The experimental data are obtained during the running at the velocity of 137 km/h in a current line and are used to define, depending on the limit values of the dispersion intervals of RMS accelerations, the normal operating area, and the failure area of the damper. Experimental data are also the basis for validating the results of numerical simulations. As for the two models of vehicle track systems for simulating the vibrations response of the bogie, both are complex, of rigid-flexible coupled type. These models consider the relevant vertical vibration modes of the car body–bounce, pitch, and the first vertical bending, the vertical vibration bounce and pitch of the bogies, the vertical displacements of the wheels and rails, and the wheel-track contact elasticity. The first model, with 15 degrees of freedom, is a reference model used to simulate the bogie’s response to the vertical vibrations for the normal operating condition of the dampers in the primary suspension. The second model is used to simulate the vibration response of the bogie to the failure of a damper in the primary suspension. The failure of a damper is represented in the model through different damping constants of the suspension of the four wheels of the bogie; asymmetric damping of the primary suspension [55]. This model is obtained by extending the first model to a model with 20 degrees of freedom while considering the roll vibrations of one of the bogies and its wheelsets, which are excited due to the damping asymmetry of the primary suspension.

Further on, the paper is structured into 5 sections. Section 2 contains a detailed presentation of the first model of the vehicle–track system for the simulation of the response to the bogie’s vertical vibrations, the motion equations, and an original method to synthesize the track irregularities previously applied, both for the synthesis of the track vertical irregularities and for the track horizontal irregularities [56,57,58]. Section 3 introduces the architecture of the measurement system for the bogie vertical accelerations in a passenger vehicle in current line traffic and the analysis of the recorded data. Section 4 examines the response to vibrations of the bogie, based on both the measured RMS accelerations and the simulated RMS accelerations. Section 5 includes the bogie-track system model used to simulate the response to bogie vibrations to failure of a primary suspension damper, the corresponding motion equations of the vehicle-track system, and the results of the numerical simulations. Section 6 is dedicated to the conclusions of the paper. 

## 2. The Vehicle–Track System Model for Simulating the Bogie’s Response to Vertical Vibrations

### 2.1. Description of the Vehicle–Track System Model

Figure 1 features the vehicle–track system model used to simulate the response to vertical vibrations of a two-axle bogie running at a constant velocity on a track with vertical irregularities. 

The vehicle model is a rigid–flexible coupled type, comprising the vehicle carbody, the two bogies, and four wheelsets, interconnected through Kelvin–Voigt systems. The vehicle carbody is represented by a free-free equivalent beam, with a constant section and uniformly distributed mass, Euler–Bernoulli type. This model allows to take into consideration the rigid vertical vibration modes of the carbody–bounce *z_c_*, pitch θ*_c_* and the first vertical bending mode. Because the bogies and wheelsets have reduced elastic deformations, they are represented by rigid bodies. The vibration modes of the two bogies in a vertical plan are the bounce *z_bi_* and pitch θ*_bi_*, with *i* = 1, 2. The wheelsets can perform a translation movement in the vertical direction—bounce *z_wj,j_*_+1_, with *j* = 2*i* − 1, specifying that each bogie *i* is equipped with the wheelsets *j* and *j* + 1.

The bogie is connected to the carbody through the secondary suspension that is modelled via a Kelvin–Voigt system, with the elastic constant 2*k_c_* and the damping constant 2*c_c_*. Each wheelset is connected to the bogie chassis through a Kelvin–Voigt system, with the elastic constant 2*k_b_* and the damping constant 2*c_b_*. The primary suspension of the bogie is modelled by two Kelvin–Voigt systems corresponding to the wheelset of a bogie. 

The coupling effects between the wheels due to the propagation of the bending waves through the rails are neglected, and an equivalent model with concentrated parameters is adopted for the track. Against each wheelset *j* and *j* + 1, respectively, of the bogie *i*, the track is represented with a one-degree freedom system on the vertical direction, the corresponding displacement being *z_rj,j+_*_1_, with *j* = 2*i* − 1 and *i* = 1, 2. The equivalent model of the track has the mass *m_r_*, elastic constant 2*k_r_* and the damping constant 2*c_r_*. The vertical irregularities of the track η*_j,_*_(*j*+1)_ are considered equal. 

By introducing elastic elements with a linear stiffness characteristic of 2*k_H_* for each wheel-rail pair, the elasticity of the wheel-rail contact is taken into account.

### 2.2. Synthesizing of the Vertical Track Irregularities

The method to synthesize the vertical track irregularities [56,57] relies on the power spectral density of the track irregularities, as described by ORE B176 [59] and the specifications stipulated by the UIC 518 Leaflet [60] concerning the track geometry quality in the testing and homologation of the railway vehicles from the perspective of the dynamic behaviour. According to this method, the vertical track irregularities are described via a pseudo-random function.
(1)$$\eta_{j,j+1}(x_{j,j+1})=f(x_{j,j+1})\sum_{k=0}^{N}U_{k}\cos(\Omega_{k}x_{j,j+1}+\varphi_{k}),\;\mathrm{for}\;x_{j,j+1}\;>\;0,$$
where *x_j,j_*_+1_, with *j* = 2*i* − 1 and *i* = 1, 2, depending on the wheelset position,
$$\begin{array}{c} x_{j}=x;\;x_{j+1}=x-2a_{b},\;\mathrm{for}\;i=1\\ x_{j}=x-2a_{c};\;x_{j+1}=x-2a_{b}-2a_{c},\;\mathrm{for}\;i=2, \end{array}$$
for *x* = *Vt*, where *V* is the vehicle velocity.

*U_k_* is the amplitude of the spectral component‚ *k*’ corresponding to the wavelength number Ω*_k_*, where
Ω*_k_* = Ω_0_
*+ k*ΔΩ, for *k* = 0, 1, 2, ... *N*, 
with *N* + 1 number of spectral components and ΔΩ = Ω_0_ − Ω*_N_*, where
$$\Omega_{0}=\frac{2\pi }{\Lambda_{\max}},\;\Omega_{N}=\frac{2\pi }{\Lambda_{\min}},$$
Λ_min_ stands for the minimum wavelength, while Λ_max_ is the maximum wavelength. 

The amplitude of the spectral component‚ *k*’ is in the form of
(2)$$U_{k}=\sqrt{\frac{1}{\pi }\Phi (\Omega_{k})\Delta \Omega },\;\mathrm{with}\;k=0,\;1,\;2,\;\ldots \;N,$$
and is calculated based on the power spectral density of the vertical track irregularities (see Figure 2), described as per ORE B176 [59]
(3)$$\Phi (\Omega_{k})=\frac{A_{QN1,2}\Omega_{c}^{2}}{\left(\Omega_{k}^{2}+\Omega_{r}^{2})(\Omega_{k}^{2}+\Omega_{c}^{2}\right)},$$
where Ω*_c_* = 0.8246 rad/m, Ω*_r_* = 0.0206 rad/m, and *A_QN_*_1,2_ is a constant that depends on the quality of the track. This theoretical curve of the power spectral density is representative for the average statistical properties of the European railway.

The quality constant of the track *A_QN_*_1,2_ is calculated in such a way that the standard deviation of the vertical track irregularities (σ_ηQN1, QN2_) due to the components with the wavelength ranging from Λ_1_ = 3 m to Λ_2_ = 25 m to correspond to the stipulations in UIC 518 Leaflet, depending on the track quality level, QN1 or QN2 (see Figure 3),
(4)$$A_{QN1,2}=2\pi \frac{\sigma_{\eta \mathrm{QN}1,\mathrm{QN}2}^{2}}{\Omega_{c}^{2}I_{0}},$$
with
(5)$$I_{0}=\int_{\Omega_{2}}^{\Omega_{1}}\frac{d\Omega_{k}}{\left(\Omega_{k}^{2}+\Omega_{r}^{2})(\Omega_{k}^{2}+\Omega_{c}^{2}\right)},\;\mathrm{for}\;\Omega_{1,2}=2\pi /\Lambda_{1},$$

Figure 4 shows the values of *A_QN_*_1,2_ calculated via Equation (4), where σ_ηQN1,2_ has been assigned to various values corresponding to a QN1 track quality and a QN2 track quality, according to the UIC 518 Leaflet (see Figure 3).

To provide randomness to the vertical track irregularities, uniform random distribution with values from –π to +π was selected for the phase shift φ*_k_* of the spectral component‚ *k*’.

The function *f*(*x_j_*_,*j*+1_) is a smoothing function applied on the distance *L*_0_, in the form of
(6)$$f(x_{j,j+1})=\left[6{\left(\frac{x_{j,j+1}}{L_{0}}\right)}^{5}-15{\left(\frac{x_{j,j+1}}{L_{0}}\right)}^{4}+10{\left(\frac{x_{j,j+1}}{L_{0}}\right)}^{3}\right]H(L_{0}-x_{j,j+1})+H(x_{j,j+1}-L_{0})$$
where *H*(.) is the Heaviside step function. 

Figure 5 features the synthesis of the vertical track irregularities for a 2 km distance for a QN1 track quality (diagram a) and for a QN2 track quality (diagram b) for the velocity interval of 120–160 km/h. The standard deviations of the vertical track irregularities come in the following values: σ_ηQN1_ = 1.4 mm; σ_ηQN2_ = 1.7 mm (see Figure 2). For the synthesis of the vertical track irregularities, the contribution of 300 spectral components with wavelengths between Λ_min_ = 3 m and Λ_max_ = 120 m was considered. For the limit values of the wavelength intervals, the minimum excitation frequency due to track irregularities at the velocity of 120 km/h was 0.27 Hz, while the minimum frequency was 11.11 Hz. The minimum excitation frequency due to track irregularities was 0.37 Hz for the velocity of 160 km/h, and the maximum frequency was 14.81 Hz. These frequency intervals cover the domain in which the eigenfrequencies of the vertical vibrations of the railway vehicle were to be found.

### 2.3. The Motion Equations for the Vehicle–Track System

The vertical motions of the vehicle are described by the equations of the rigid vibration modes of the carbody and bogies–bounce and pitch, the equation of the first bending eigenmode of the carbody and the equations of the vertical displacements of the wheelsets. 

To write the motion equation of the carbody, the following parameters are being looked at: *m_c_*—carbody mass; *L_c_*—carbody length; μ—structural damping coefficient; *EI*—bending module, where *E* is the elasticity longitudinal modulus, and *I* is the inertia moment of the transversal section. 

The carbody displacement *w_c_*(*x,t*) is the result of the overlapping of the three vibration modes—bounce, pitch, and the first vertical bending mode,
(7)$$w_{c}(x,t)=z_{c}(t)+\left(x-\frac{L_{c}}{2}\right)\theta_{c}(t)+X_{c}(x)T_{c}(t),\;$$
where *T_c_*(*t*) is the coordinate of the carbody vertical bending, with *X_c_*(*x*) as its eigenfunction
(8)$$X_{c}(x)=\sin\beta x+\sinh\beta x-\frac{\sin\beta L-\sinh\beta L}{\cos\beta L-\cosh\beta L}(\cos\beta x+\cosh\beta x),$$
with
(9)$$\beta =\sqrt[{4}]{\omega_{c}^{2}\rho_{c}/(EI)}\;\mathrm{and}\;\cos\beta L_{c}\cosh\beta L_{c}-1=0,\;$$
where ω*_c_* is the eigen pulsation of the carbody vertical bending and ρ*_c_* = *m_c_*/*L_c_*.

The carbody motion equation has the general form as below
(10)$$EI\frac{\partial^{4}w_{c}(x,t)}{\partial x^{4}}+\mu I\frac{\partial^{5}w_{c}(x,t)}{\partial x^{4}\partial t}+\rho_{c}\frac{\partial^{2}w_{c}(x,t)}{\partial t^{2}}=\sum_{i=1}^{2}F_{ci}\delta (x-l_{i}),$$
where δ(.) is Dirac delta function, distances *l_i_* (for *i* = 1, 2) the position of the carbody support points on the secondary suspension, and is expressed as a function of the carbody length *L_c_* and the carbody wheelbase 2*a_c_*, namely *l_i_* = *L_c_*/2 ± *a_c_*, while *F_ci_* represents the force due to the secondary suspension of the bogie *i*,
(11)$$F_{ci}=-2c_{c}\left(\frac{\partial w_{c}(l_{i},t)}{\partial t}-{\dot{z}}_{bi}\right)-2k_{c}\left[w_{c}(l_{i},t)-z_{bi}\right].$$

With the help of the modal analysis method and given the orthogonality property of the eigenfunction of the carbody vertical bending, the motion Equation (10) converts into three second order differential equations with ordinary derivatives, which describe the three vibration modes of the carbody—bounce, pitch, and the vertical bending,
(12)$$m_{c}{\ddot{z}}_{c}=\sum_{i=1}^{2}F_{ci},$$
(13)$$J_{c\theta }{\ddot{\theta }}_{c}=\sum_{i=1}^{2}F_{ci}\left(l_{i}-\frac{L_{c}}{2}\right),$$
(14)$$m_{mc}{\ddot{T}}_{c}+c_{mc}{\dot{T}}_{c}+k_{mc}T_{c}=\sum_{i=1}^{2}F_{ci}X_{c}(l_{i}),$$
with the following notations: *J_c_*_θ_—carbody pitch moment of inertia, *m_mc_*—carbody modal mass, *c_mc_*—carbody modal damping and *k_mc_*—carbody modal stiffness. The modal mass, modal damping and the modal stiffness are defined as follows: (15)$$m_{mc}=\rho_{c}\int_{0}^{L}X_{c}^{2}dx,\;c_{mc}=\mu I\int_{0}^{L}{\left(\frac{d^{2}X_{c}}{dx^{2}}\right)}^{2}dx,\;k_{mc}=EI\int_{0}^{L}{\left(\frac{d^{2}X_{c}}{dx^{2}}\right)}^{2}dx.$$

Based on the symmetry properties of the eigenfunction *X_c_*(*x*), the following notation is adopted
(16)$$X_{c}(l_{1})=X_{c}(l_{2})=\varepsilon$$

After processing, the carbody motion Equations (12)–(14) are in the form of
(17)$$m_{c}{\ddot{z}}_{c}+2c_{c}[2{\dot{z}}_{c}+2\varepsilon {\dot{T}}_{c}-({\dot{z}}_{b1}+{\dot{z}}_{b2})]+2k_{c}[2z_{c}+2\varepsilon T_{c}-(z_{b1}+z_{b2})]=0$$
(18)$$J_{c\theta }{\ddot{\theta }}_{c}+2c_{c}a_{c}[2a_{c}{\dot{\theta }}_{c}-({\dot{z}}_{b1}-{\dot{z}}_{b2})]+2k_{c}a_{c}[2a_{c}\theta_{c}-(z_{b1}-z_{b2})]=0\;$$
(19)$$\begin{array}{l} m_{mc}{\ddot{T}}_{c}+c_{mc}{\dot{T}}_{c}+k_{mc}T_{c}+\\ +2c_{zc}\varepsilon [2{\dot{z}}_{c}+2\varepsilon {\dot{T}}_{c}-({\dot{z}}_{b1}+{\dot{z}}_{b2})]+2k_{zc}\varepsilon [2z_{c}+2\varepsilon T_{c}-(z_{b1}+z_{b2})]=0 \end{array}$$

The motion equations of the two bogies are as below
-Bounce motion equation for the bogie *i*(20)$$m_{b}{\ddot{z}}_{bi}=\sum_{j=2i-1}^{2i}F_{bj}-F_{ci},\;\mathrm{with}\;i=1,\;2\;$$-Pitch motion equation for the bogie *i*
(21)$$J_{b\theta }{\ddot{\theta }}_{bi}=a_{b}\sum_{j=2i-1}^{2i}{\left(-1\right)}^{j+1}F_{bj},\;\mathrm{with}\;i=1,\;2,$$
where *m_b_* is the bogie mass, *J_b_*_θ_ is the bogie pitch moment of inertia, 2*a_b_* is the bogie wheelbase and *F_b_* is the force due to primary suspension. 

The forces due to the suspension corresponding to the wheelsets *j* and *j* + 1 are in the form of
(22)$$F_{bj,(j+1)}=-2c_{b}({\dot{z}}_{bi}\pm a_{b}{\dot{\theta }}_{bi}-{\dot{z}}_{wj,(j+1)})-2k_{b}(z_{bi}\pm a_{b}\theta_{bi}-z_{wj,(j+1)}),\;$$
with *j* = 2*i* − 1 and *i* = 1, 2.

The bounce and pitch bogie equations write as
(23)$$\begin{array}{l} m_{b}{\ddot{z}}_{bi}+2c_{b}[2{\dot{z}}_{bi}-({\dot{z}}_{wj}+{\dot{z}}_{w(j+1)})]+2k_{b}[2z_{bi}-(z_{wj}+z_{w(j+1)})]+\\ +2c_{c}({\dot{z}}_{bi}-{\dot{z}}_{c}\mp a_{c}{\dot{\theta }}_{c}-\varepsilon {\dot{T}}_{c})+2k_{c}(z_{bi}-z_{c}\mp a_{c}\theta_{c}-\varepsilon T_{c})=0\; \end{array}$$
(24)$$J_{b\theta }{\ddot{\theta }}_{bi}+2c_{b}a_{b}[2a_{b}{\dot{\theta }}_{bi}-({\dot{z}}_{wj}-{\dot{z}}_{w(j+1)})]+2k_{b}a_{b}[2a_{b}\theta_{bi}-(z_{wj}-z_{w(j+1)})]=0.\;$$

The equations of the motions in the vertical direction of the wheelsets *j* and *j* + 1 are
(25)$$m_{w}{\ddot{z}}_{wj,(j+1)}=2Q_{j,(j+1)}-F_{bj,(j+1)},\;$$
where *Q_j_*_,(*j*+1)_ stand for the dynamic wheel-rail contact forces. To calculate the dynamic forces, the hypothesis of the linear wheel-rail hertzian contact is adopted,
(26)$$Q_{j,(j+1)}=-k_{H}[z_{wj,(j+1)}-z_{rj,(j+1)}-\eta_{j,(j+1)}],\;\mathrm{with}\;j=2i-1\;\mathrm{and}\;i=1,\;2.$$

After processing, Equation (25) changes to
(27)$$\begin{array}{l} m_{w}{\ddot{z}}_{wj,(j+1)}+2c_{b}({\dot{z}}_{wj.(j+1)}-{\dot{z}}_{bi}\mp a_{b}{\dot{\theta }}_{bi})+2k_{z}(z_{wj,(j+1)}-z_{bi}\mp a_{b}\theta_{bi})+\\ +2k_{H}(z_{wj,(j+1)}-z_{rj,(j+1)}-\eta_{j,(j+1)})=0\; \end{array}$$

The equations of the vertical rail displacements against the wheelsets *j* and (*j* + 1) are
(28)$$m_{r}{\ddot{z}}_{rj,(j+1)}=F_{rj,(j+1)}-2Q_{j,(j+1)},\;$$
where
(29)$$F_{rj,(j+1)}=-2c_{r}{\dot{z}}_{rj,(j+1)}-2k_{r}z_{rj,(j+1)},\;\mathrm{with}\;j=2i-1\;\mathrm{and}\;i=1,\;2.\;$$

Upon replacement, Equation (28) becomes
(30)$$m_{r}{\ddot{z}}_{rj,(j+1)}+2c_{r}{\dot{z}}_{rj,(j+1)}+2k_{r}z_{rj,(j+1)}+2k_{H}(z_{rj,(j+1)}-z_{rj,(j+1)}+\eta_{j,(j+1)})=0.\;$$

The motion Equations (17)–(19), (23), (24), (27) and (30) form a system of 15 s order differential equations, in which the condition variables, displacements, and velocities are introduced, such as:(31)$$q_{2s-1}=p_{s},\;q_{2s}={\dot{p}}_{s},\;\mathrm{for}\;s=1\ldots \;15.$$

Hence, it results a system of 30 first order differential equations that can be written in matrix form,
(32)$$\dot{q}=Aq+B,$$
where **q** is the vector of the condition variables, **A** is the system matrix and **B**—the vector of the non-homogeneous terms. The system of Equation (32) is solved through numerical integration.

## 3. The Measurement of the Bogie Vertical Accelerations

This section describes the architecture of the measurement chain for the bogie vertical accelerations of a passenger vehicle during circulation on a current line at the maximum velocity of 160 km/h, on different track sections in alignment and crosslevel. The vehicle is equipped with Minden–Deutz bogies (see Figure 6) with a maximum velocity of 140 km/h. The values of the RMS accelerations for each measurement section are also featured.

The architecture of the measurement chain for the bogie accelerations is shown in Figure 7. It comprises the components of the measurement system, acquisition, and processing of the bogie accelerations, namely four Brüel and Kjær 4514 accelerometers and the ensemble including the chassis NI cDAQ-9174 for data acquisition and the NI 9234 module for acquisition and synthesizing the data from accelerometers. To monitor and record the vehicle velocity, the receiver GPS NL-602U was used [61,62,63,64].

The four accelerometers are mounted on one side of the front bogie, according to the direction of travel, with one accelerometer on each axle box and one accelerometer on the bogie frame against the suspension of each wheelset. The position of the four accelerometers is shown in Figure 8.

Recordings of accelerations during the circulation at the constant velocity of 137 km/h were made, for 17 measurement sequences of 20 s duration, with 2048 samples per second. For instance, Figure 9 presents the accelerations recorded by the four accelerometers on a measuring sequence.

The diagrams in Figure 10 show the values of the RMS accelerations for 17 measurement sequences. For the axle boxes, the RMS accelerations were found in the 1.27... 2.14 m/s^2^ interval and 1.19... 2.08 m/s^2^, respectively. In the two measurement points on the bogie frame, the RMS accelerations had values ranging from 0.65 m/s^2^ to 1.05 m/s^2^, respectively, between 0.65 m/s^2^ and 1.22 m/s^2^. The dispersion of the RMS accelerations values can be explained via the amplitudes of the track irregularities. The RMS accelerations for the axle box were noticed to be twice as big as the RMS accelerations on the bogie frame. 

## 4. The Analysis of the Bogie to Vibrations Response Based on the Measured and Simulated Accelerations 

In this section, the bogie accelerations derived from the numerical simulations developed based on the mechanical model of the vehicle–track system featured in Section 2 of the paper herein were compared with the measured accelerations. 

The values of the parameters of the numerical model of the vehicle–track system used in the numerical simulations were adopted in conformity with the characteristics of the passenger vehicle for which the bogie vertical accelerations were measured (see Table 1). The synthesis of the vertical track irregularities was conducted in accordance with the methodology introduced in Section 2.3, for a 2 km distance, for the standard deviations of the vertical irregularities corresponding to the QN1 track quality and QN2 track quality for the velocity interval of 120–160 km/h (see Figure 5).

Figure 11 and Figure 12 show the accelerations derived from numerical simulations in four reference bogie points, marked on the bogie 1 model in Figure 1, where such points are homologous with the measurement points. The correspondence between these points is the following: point *B*_1_—accelerometer 1, point *B*_2_—accelerometer 2, point *B_w_*_1_—accelerometer 3, point *B_w_*_2_—accelerometer 4. 

To be compared with the simulated accelerations, the measured accelerations were band-pass filtered in the 0.31–12.68 Hz frequency domain. These limits of the frequency domain corresponded to the limits of the interval of wavelengths ranging from 3 to 120 m.

Figure 13 shows the values of the measured RMS accelerations and the values of RMS accelerations obtained by numerical simulations to circulation on a QN1 track quality and on a QN2 track quality. The values of the RMS accelerations measured on the bogie frame against the front wheelset suspension (diagram a) were dispersed between 1.38 m/s^2^ and 1.83 m/s^2^, with an average value of 1.56 m/s^2^, and against the suspension of the second wheelset (diagram b) between 1.38 m/s^2^ and 2.19 m/s^2^, with an average value of 1.69 m/s^2^. At the axle box of wheelset 1 (diagram c), the measured RMS accelerations were found between the minimum value of 1.09 m/s^2^ and the maximum value of 1.90 m/s^2^, with the average value of 1.44 m/s^2^. At the axle box of wheelset 2 (diagram d), the measured RMS accelerations were between 1.13 and 1.94 m/s^2^, with an average value of 1.48 m/s^2^. The limits to the dispersion intervals of the RMS accelerations values were considered to define the normal operating area of the damper. Consequently, the failure area of the damper can be described as being delineated by the superior limit of the dispersion interval of the RMS accelerations. In all four cases, the values of the RMS accelerations simulated for the two track quality levels were within the dispersion intervals of the measured RMS accelerations. Under these circumstances, the average of the measured RMS accelerations was chosen as a reference. Table 2 features the percent differences between the RMS accelerations from the numerical simulations and the average between the measured RMS accelerations. It is noted that the difference between the value of RMS acceleration derived from numerical simulations for a QN1 track quality and the average of the measured RMS accelerations can be as high as 20%. Instead, for the RMS accelerations simulated at the circulation on a QN2 track quality, the difference towards the average of the measured RMS accelerations does not exceed 9%, a value considered acceptable to validate the proposed model of the vehicle–track system for simulating the bogie response to vertical vibrations. 

## 5. Analysis of the Bogie Response to the Failure of a Damper in the Primary Suspension

### 5.1. The Model of the Bogie–Track System

Figure 14 shows the bogie-track system model used in this section to simulate the bogie response to the failure of a damper of the primary suspension. This model integrates itself in the model of the vehicle-track system in Figure 1, replacing the model of bogie 1. 

The failure of a damper is represented in the bogie model via different damping constants of the four Kelvin–Voigt systems through which the suspensions against each wheel are modelled (*c_b_*_11_ ≠ *c_b_*_12_ ≠ *c_b_*_21_ ≠ *c_b_*_22_ ≠ *c_b_*, where *c_b_* stands for the reference damping constant).

The asymmetry of the suspension damping results into the excitation of the bogie roll vibrations φ*_b_*_1_ that interferes with its vertical vibrations of bounce *z_b_*_1_ and pitch θ*_b_*_1_. Still an effect of the suspension asymmetry, the wheelsets of the front bogie will also show a rotation–roll motion φ*_w_*_1, 2_ (see Figure 15), along with a bounce motion *z_w_*_1,2_. As a result, the vertical displacements of the two wheels of the wheelset are different, as seen below
(33)$$z_{w11,12}=z_{w1}\mp e\varphi_{w1},\;z_{w21,22}=z_{w2}\mp e\varphi_{w2}.$$
where *e* represents the semi-distance between the wheel-rail contact points for the wheelset in the median position on the track. 

Similarly, the vertical displacements of the rails against the wheels of the two wheelsets are different, *z_r_*_11_ ≠ *z_r_*_12_ and *z_r_*_21_ ≠ *z_r_*_22_.

The model in Figure 14 has four reference bogie points, homologous to the measurement points (see Figure 7 and Figure 8). Two of these points, noted with *B*_11_ and *B*_21_, are located against the supporting points of the chassis on the suspension corresponding to the wheels 11 and 21, as being the counterparts of the points where the accelerometers 1 and 2 are mounted. The other two points, noted with *B_w_*_11_ and *B_w_*_21_, are situated against the axle boxes of the wheels 11 and 21, as being the counterparts of the points where the accelerometers 3 and 4 are mounted.

The vertical motions of the vehicle–track system are described by the carbody equations of bounce, pitch, and bending (Equations (17)–(19)), the bogie 2 bounce and pitch equations (Equations (23) and (24), for *i* = 2 and *j* = 3), the equations of the vertical displacements of the wheelsets 3 and 4 (Equation (25), for *j* = 3), the equations of the vertical displacements of the rails against the wheelsets 3 and 4 (Equation (30), for *j* = 3), to which the motion equations corresponding to the bogie-track system in Figure 14 are added, as such:-Equation of bogie 1 bounce,(34)$$m_{b}{\ddot{z}}_{b1}=\sum_{j=1}^{2}\sum_{k=1}^{2}F_{bjk}-F_{c1};\;$$-Equation of bogie 1 pitch,(35)$$J_{b\theta }{\ddot{\theta }}_{b1}=a_{b}\sum_{j=1}^{2}\sum_{k=1}^{2}{\left(-1\right)}^{j+1}F_{bjk};\;$$-Equation of bogie 1 roll,
(36)$$J_{b\varphi }{\ddot{\varphi }}_{b1}=-b\sum_{j=1}^{2}\sum_{k=1}^{2}{\left(-1\right)}^{k+1}F_{bjk}$$
where *J_b_*_φ_ is the inertia moment to bogie roll, 2*b* is the lateral base of the primary suspension, while *F_bjk_* (see Figure 15) represents the forces in the suspension corresponding to each wheel of bogie 1 (with *j* = 1, 2 and *k* = 1, 2) in the form of
(37)$$F_{b11,12}=-c_{11,12}({\dot{z}}_{b1}+a_{b}{\dot{\theta }}_{b1}\mp b{\dot{\varphi }}_{b1}-{\dot{z}}_{w11,12})-k_{b}(z_{b1}+a_{b}\theta_{b1}\mp b\varphi_{b1}-z_{w11,12}),\;$$
(38)$$F_{b21,22}=-c_{21,22}({\dot{z}}_{b1}-a_{b}{\dot{\theta }}_{b1}\mp b{\dot{\varphi }}_{b1}-{\dot{z}}_{w21,22})-k_{b}(z_{b1}-a_{b}\theta_{b1}\mp b\varphi_{b1}-z_{w21,22}).\;$$-Vertical motion equations of the wheelsets *j*, for *j* = 1, 2,
(39)$$m_{w}{\ddot{z}}_{wj}=\sum_{k=1}^{2}(Q_{jk}-F_{bjk}),\;$$
where the dynamic forces of wheel-rail contact *Q_jk_*, for *k* = 1, 2, (see Figure 15) are defined as per the equations
(40)$$Q_{11,12}=-k_{H}(z_{w11,12}-z_{r11,12}-\eta_{1}),\;$$
(41)$$Q_{21,22}=-k_{H}(z_{w21,22}-z_{r21,22}-\eta_{2}).\;$$-Roll equations of the wheelsets *j*, for *j* = 1, 2,(42)$$J_{w}{\ddot{\varphi }}_{wj}=\sum_{k=1}^{2}[{\left(-1\right)}^{k}eQ_{jk}+{\left(-1\right)}^{k+1}bF_{bjk}].\;$$-Equations of the vertical displacements of the rails against the wheelsets *j*,
(43)$$m_{w}{\ddot{z}}_{rj}=\sum_{k=1}^{2}(F_{rjk}-Q_{jk}),\;\mathrm{for}\;j=1,\;2.\;$$
where
(44)$$F_{rjk}=-c_{r}{\dot{z}}_{rjk}-k_{r}z_{rjk},\;\mathrm{for}\;k=1,\;2.\;$$

The motion equations of the system formed by bogie 1 and the track are brought into the form of:(45)$$\begin{array}{l} m_{b}{\ddot{z}}_{b1}+(c_{11}+c_{12}+c_{21}+c_{22}+2c_{c}){\dot{z}}_{b1}+2(2k_{b}+k_{c})z_{b1}+\\ +a_{b}(c_{11}+c_{12}-c_{21}-c_{22}){\dot{\theta }}_{b1}-b(c_{11}-c_{12}+c_{21}-c_{22}){\dot{\varphi }}_{b1}-\\ -(c_{11}+c_{12}){\dot{z}}_{w}{}_{1}-(c_{21}+c_{22}){\dot{z}}_{w}{}_{2}-2k_{b}(z_{w}{}_{1}+z_{w}{}_{2})+\\ +e(c_{11}-c_{12}){\dot{\varphi }}_{w1}+e(c_{21}-c_{22}){\dot{\varphi }}_{w2}-\\ -2c_{c}({\dot{z}}_{c}+a_{c}{\dot{\theta }}_{c}+\varepsilon {\dot{T}}_{c})-2k_{c}(z_{c}+a_{c}\theta_{c}+\varepsilon T_{c})=0 \end{array}$$
(46)$$\begin{array}{l} J_{b\theta }{\ddot{\theta }}_{b1}+a_{b}^{2}(c_{11}+c_{12}+c_{21}+c_{22}){\dot{\theta }}_{b1}+4a_{b}^{2}k_{b}\theta_{b}+\\ +a_{b}(c_{11}+c_{12}-c_{21}-c_{22}){\dot{z}}_{b1}-a_{b}b(c_{11}-c_{12}-c_{21}+c_{22}){\dot{\varphi }}_{b1}\\ -a_{b}(c_{11}+c_{12}){\dot{z}}_{w1}+a_{b}(c_{21}+c_{22}){\dot{z}}_{w2}-2a_{b}k_{b}(z_{w1}-z_{w2})+\\ +a_{b}e(c_{11}-c_{12}){\dot{\varphi }}_{w1}-a_{b}e(c_{21}-c_{22}){\dot{\varphi }}_{w2}=0 \end{array}$$
(47)$$\begin{array}{l} J_{b\varphi }{\ddot{\varphi }}_{b1}+b^{2}(c_{11}+c_{12}+c_{21}+c_{22}){\dot{\varphi }}_{b1}+4b^{2}k_{b}\varphi_{b1}-\\ -b(c_{11}-c_{12}+c_{21}-c_{22}){\dot{z}}_{b1}-a_{b}b(c_{11}-c_{12}-c_{21}+c_{22}){\dot{\theta }}_{b1}+\\ +b(c_{11}-c_{12}){\dot{z}}_{w1}+b(c_{21}-c_{22}){\dot{z}}_{w2}-\\ -be(c_{11}-c_{12}){\dot{\varphi }}_{w1}-be(c_{21}-c_{22}){\dot{\varphi }}_{w2}-2bek_{b}(\varphi_{w1}+\varphi_{w2})=0 \end{array}$$
(48)$$\begin{array}{l} m_{w}{\ddot{z}}_{w1}+(c_{11}+c_{12}){\dot{z}}_{w1}+2(k_{b}+k_{H})z_{w1}-e(c_{11}-c_{12}){\dot{\varphi }}_{w1}-\\ -(c_{11}+c_{12}){\dot{z}}_{b1}-2k_{b}z_{b1}-a_{b}(c_{11}+c_{12}){\dot{\theta }}_{b1}-2a_{b}k_{b}\theta_{b1}+\\ +b(c_{11}-c_{12}){\dot{\varphi }}_{b1}-k_{H}(z_{r11}+z_{r12})=2k_{H}\eta_{1} \end{array}$$
(49)$$\begin{array}{l} m_{w}{\ddot{z}}_{w2}+(c_{21}+c_{22}){\dot{z}}_{w2}+2(k_{b}+k_{H})z_{w2}-e(c_{21}-c_{22}){\dot{\varphi }}_{w2}-\\ -(c_{21}+c_{22}){\dot{z}}_{b1}-2k_{b}z_{b1}+a_{b}(c_{21}+c_{22}){\dot{\theta }}_{b1}+2a_{b}k_{b}\theta_{b1}+\\ +b(c_{21}-c_{22}){\dot{\varphi }}_{b1}-k_{H}(z_{r21}+z_{r22})=2k_{H}\eta_{2} \end{array}$$
(50)$$\begin{array}{l} J_{w\varphi }{\ddot{\varphi }}_{w1}+be(c_{11}+c_{12}){\dot{\varphi }}_{w1}+2e(bk_{b}+ek_{H})\varphi_{w1}-b(c_{11}-c_{12}){\dot{z}}_{w1}+\\ +b(c_{11}-c_{12}){\dot{z}}_{b1}+a_{b}b(c_{11}-c_{12}){\dot{\theta }}_{b1}-b^{2}(c_{11}+c_{12}){\dot{\varphi }}_{b1}-2b^{2}k_{b}\varphi_{b1}+\\ +ek_{H}(z_{r11}-z_{r12})=0 \end{array}$$
(51)$$\begin{array}{l} J_{w\varphi }{\ddot{\varphi }}_{w2}+be(c_{21}+c_{22}){\dot{\varphi }}_{w2}+2e(bk_{b}+ek_{H})\varphi_{w2}-b(c_{21}-c_{22}){\dot{z}}_{w2}+\\ +b(c_{21}-c_{22}){\dot{z}}_{b1}-a_{b}b(c_{21}-c_{22}){\dot{\theta }}_{b1}-b^{2}(c_{21}+c_{22}){\dot{\varphi }}_{b1}-2b^{2}k_{b}\varphi_{b1}+\\ +ek_{H}(z_{r21}-z_{r22})=0 \end{array}$$
(52)$$m_{r}{\ddot{z}}_{r11,12}+2c_{r}{\dot{z}}_{r11,12}+(k_{r}+k_{H})z_{r11,12}-k_{H}z_{w1}\pm ek_{H}\varphi_{w1}=-k_{H}\eta_{1}$$
(53)$$m_{r}{\ddot{z}}_{r21,22}+2c_{r}{\dot{z}}_{r21,22}+(k_{r}+k_{H})z_{r21,22}-k_{H}z_{w2}\pm ek_{H}\varphi_{21}=-k_{H}\eta_{2}$$

The motion equations of the vehicle–track system make up a system of 20 s order differential equations, in which the condition variables, displacements and the velocities are introduced in the form of
(54)$$q_{2s-1}=p_{s},\;q_{2s}={\dot{p}}_{s},\;\mathrm{for}\;s=1\ldots \;20,$$

The result is a system of 40 first-order differential equations, in a matrix form, as in Equation (32), which can be solved via numerical integration. 

### 5.2. The Results of the Numerical Simulations

This section analyses the vibration response of the bogie 1, represented by the model in Figure 14. To this purpose, the results from the numerical simulations were used concerning the RMS accelerations at a velocity of 137 km/h in the bogie reference points *B*_11_ and *B*_21_, along with *B_w_*_11_ and *B_w_*_21_, points homologous to the measurement points (see Section 3). The values of the parameters of the numerical model of the vehicle–track system used in the numerical simulations are shown in Table 1, to which the following parameters of numerical simulation were added: the lateral base of the suspension, 2*b* = 2 m, and the semi-distance between the wheel-rail contact points, 2*e* = 1.5 m. The vertical track irregularities were synthesized for a 2 km distance for a QN2 track quality (see Figure 5, diagram b), while considering the standard deviations of the vertical track irregularities corresponding to the velocity interval of 120–160 km/h. 

The failure of the damper was simulated through various reduction degrees of the damping constant compared to the reference value *c_b_* (see Table 1). Four analysis cases are being looked into, as in Table 3, where each of them corresponds to the failure of one of the four dampers.

Based on the RMS accelerations in the reference points *B*_11_ and *B*_21_, shown in the diagrams in Figure 16, Figure 17, Figure 18 and Figure 19, the vibration response of the bogie chassis for the four failure cases was analysed. The diagrams show the areas of normal operating of the damper, identified in Section 4 (see Figure 13). The higher values of the RMS acceleration than the superior limit of the normal operating area will indicate a faulty operation of the damper.

A first observation regards the fact that irrespective of the failure case, an increase of the RMS accelerations in the two bogie reference points are recorded. This increase is more or less significant, depending on the position of the damper failing and the failure degree. The important increments of the RMS accelerations are recorded in the reference point of the bogie chassis located against the faulty damper or in the diagonally-opposite point to the faulty damper, for high degrees of reduction in the damping constant. For instance, upon the failure of the damper in the wheel 11 suspension (Figure 16) or the wheel 22 (Figure 19), should the damping constant be reduced by more than 50%, the RMS acceleration in point *B*_11_ records an increase of up to 41% and up to 30%, respectively. Under such circumstances, the values of the RMS acceleration no longer belong to the area indicating the normal operating area of the damper. Similarly, for a damper failure in the suspension of wheel 12 (Figure 17) or wheel 21 (Figure 18), for reductions higher than 75% of the damping constant, the values of the RMS acceleration in the point *B*_21_ exceed the superior limit of the normal operating area for the damper, shifting to the failure area. 

The observations above show that there is a premise concerning the possibilities of fault detection in one of the dampers in the vehicle’s primary suspension, based on the analysis of the bogie response to vibrations, expressed as the RMS acceleration measured only in two reference points of the bogie chassis. 

The possibility of detecting the condition of the dampers of the primary suspension of the vehicle based on the RMS accelerations measured on the axle boxes is further investigated. For this purpose, the diagrams in Figure 20 are used, showing the RMS accelerations of the wheels 11 and 21, calculated in the points *B_w_*_11_ and *B_w_*_21_ at the velocity of 137 km/h for the four failure cases in Table 3. In this case, small variations of the RMS accelerations are obtained by reducing the damping constant; the RMS accelerations remain in the normal operating area of the damper, namely 1.09... 1.90 m/s^2^ or 1.13... 1.94 m/s^2^, defined in Figure 13. Opposite trends of the RMS accelerations in the two wheels upon the reduction in the damping constant are highlighted—the RMS acceleration decreases in the wheel against the faulty damper and in the wheel opposite to the faulty damper and increases in the other wheel. For instance, when the damper fails in the suspension of the wheel 11 (diagram a) or in the suspension of the wheel 22 (diagram d), the RMS acceleration of the wheel 11 decreases, while the RMS acceleration of the wheel 21 increases. Should the damper in the suspension of the wheel 12 (diagram b) or in the suspension of the wheel 21 (diagram c) fail, then the RMS acceleration of the wheel 21 goes down, whereas the RMS acceleration of the wheel 11 goes up. 

## 6. Conclusions

This paper investigated the possibility of fault detection of a primary suspension damper of the railway vehicle based on the analysis of the response to the vertical vibrations of the bogie. For this purpose, both experimental data and results obtained by numerical simulations were used, regarding the Root Mean Square (RMS) accelerations measured/simulated in four reference points of the bogie—two points located on the chassis against the suspension, and two points against the axle boxes. 

The analysis of the bogie response to vibrations based on the measured accelerations has pointed out to the dispersion interval of the RMS accelerations and observations concerning the ratio between the RMS accelerations measured of the axle box and the RMS accelerations measured of the bogie chassis. Regarding the first observation, it was highlighted that the RMS accelerations were dispersed in a relatively large interval, which was explained through the variability of the amplitudes of the vertical track irregularities. In this context, the notion of the normal operating area of the damper was introduced, defined by the limits of the interval of dispersion for the RMS acceleration in the four reference points of the bogie, and the notion of failure area of the damper, delineated inferiorly by the superior limit of the interval of dispersion of the RMS accelerations. Regarding the ratio of RMS accelerations measured in the four reference points of the bogie, it was shown that the RMS acceleration on the axle box was about two times higher than the RMS acceleration on the bogie frame. 

The response to vibrations in the four bogie reference points to a damper failure was examined for four cases, where each case corresponds to the failure of one of four bogie dampers. The damper failure was simulated via four degrees of reduction in the reference damping constant, namely 25%, 50%, 75%, and 100%. The results on the RMS accelerations in the reference points located on the bogie chassis show an amplification of the bogie response to vibrations upon the failure of any of the four dampers. The significant increases in the RMS acceleration, which is able to indicate the damper failure, are recorded for reductions in the damping constant by at least 50% in the reference point of the bogie chassis located against the faulty damper or in the reference point, diagonally opposite to the faulty damper. As for the RMS accelerations in the reference points against the axle boxes, they record very small variations at the simulation of damper failure, and they remain in the normal operating area.

The latest conclusions confirm that it is very likely to develop a fault detection method in a damper of the primary suspension, based on the analysis of the bogie response to vibrations, expressed in the form of RMS acceleration. The strength of this method lies in the fact that it is possible to detect a failure in any of the four dampers of the primary suspension, corresponding to a bogie based on the RMS accelerations measured in only two reference points of the bogie chassis. 

## Figures and Tables

**Figure 1 sensors-22-03290-f001:**
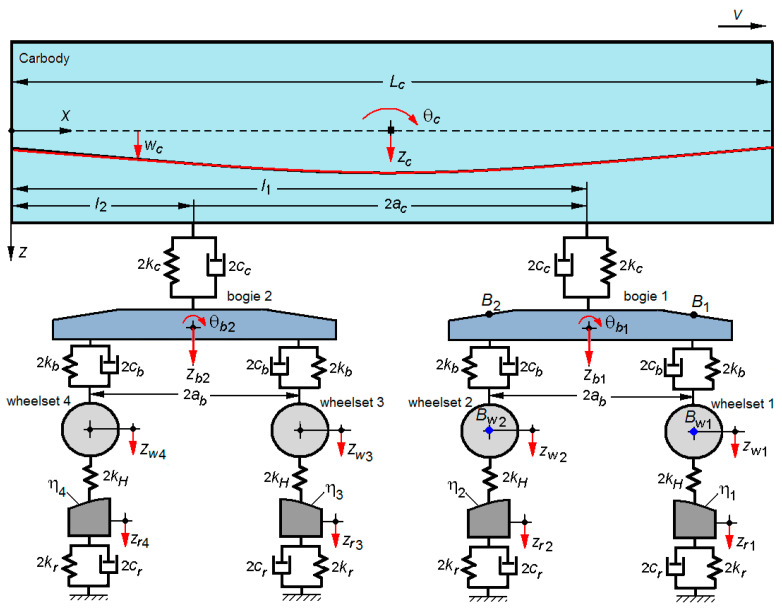
The mechanical model of the vehicle–track system.

**Figure 2 sensors-22-03290-f002:**
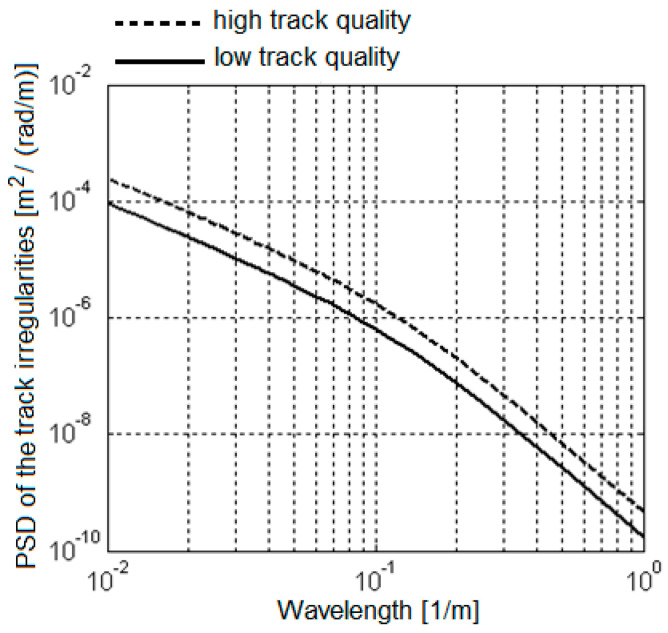
The power spectral density of the vertical track irregularities.

**Figure 3 sensors-22-03290-f003:**
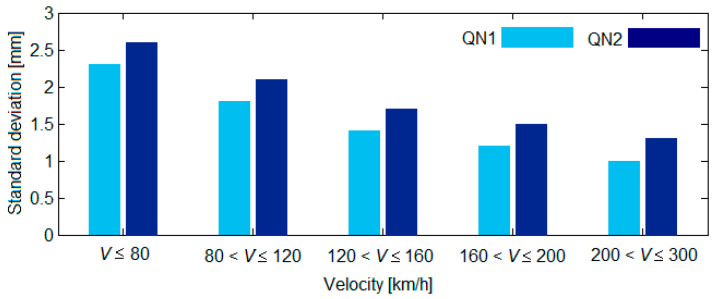
Standard deviations for vertical track irregularities.

**Figure 4 sensors-22-03290-f004:**
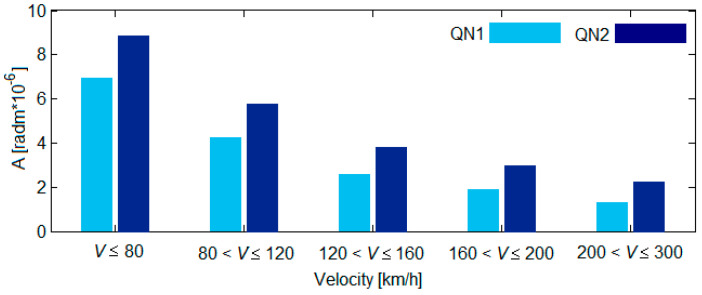
The quality constant of the track.

**Figure 5 sensors-22-03290-f005:**
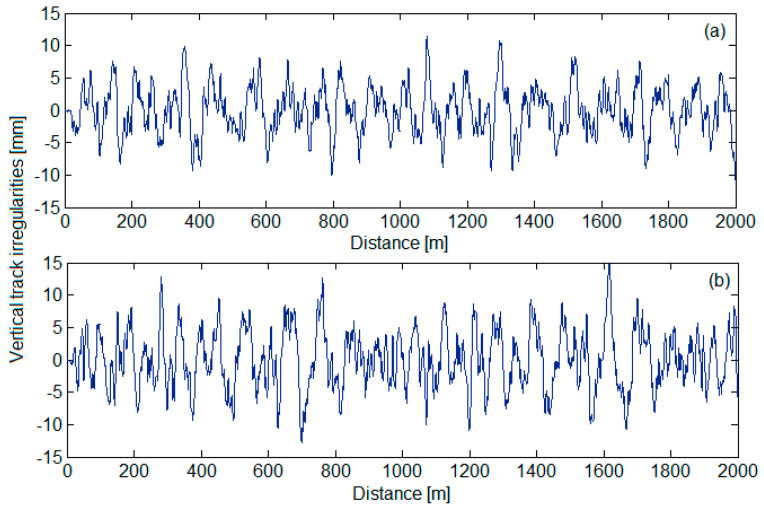
Synthesis of the vertical track irregularities.

**Figure 6 sensors-22-03290-f006:**
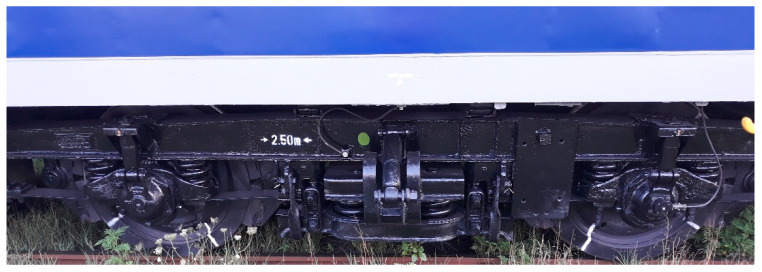
Minden–Deutz bogie.

**Figure 7 sensors-22-03290-f007:**
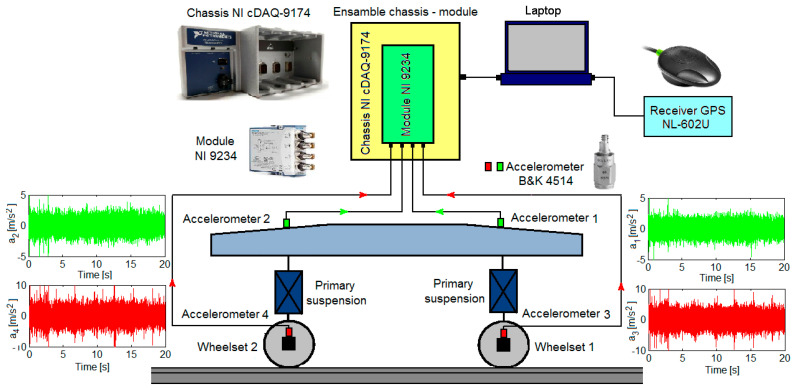
Architecture of the measurement chain of the bogie accelerations and the vehicle velocity.

**Figure 8 sensors-22-03290-f008:**
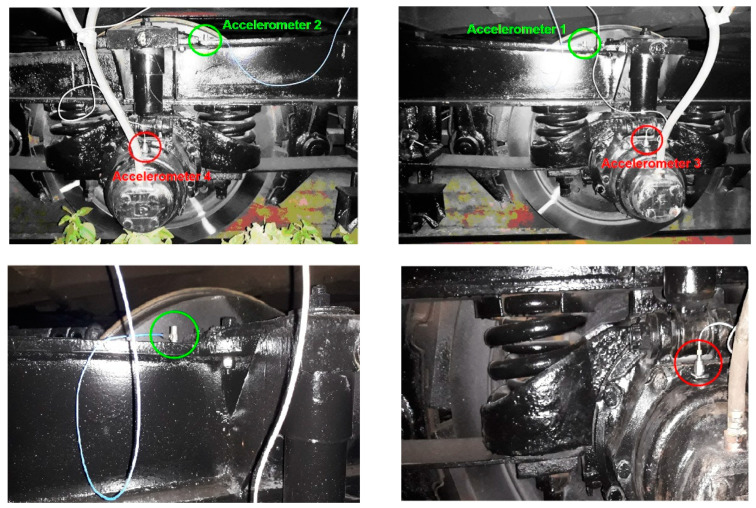
Mounting accelerometers on the bogie frame and the axle boxes.

**Figure 9 sensors-22-03290-f009:**
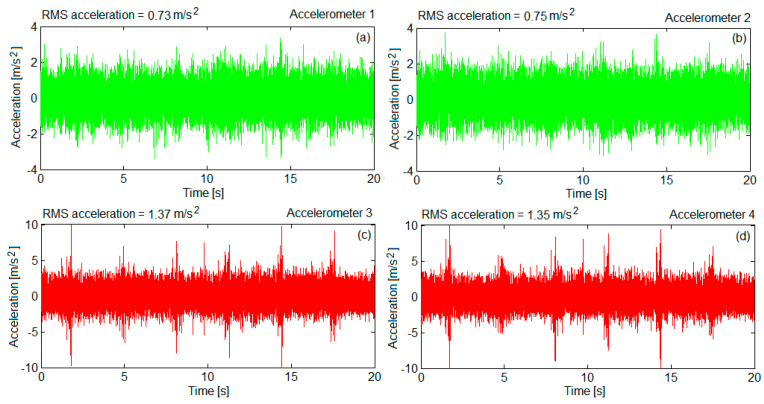
Example of accelerations recorded on a measurement sequence at the velocity of 137 km/h: (**a**) on the bogie frame above the wheelset 1; (**b**) on the bogie frame above the wheelset 2; (**c**) for the axle box of wheelset 1; (**d**) for the axle box of wheelset 2.

**Figure 10 sensors-22-03290-f010:**
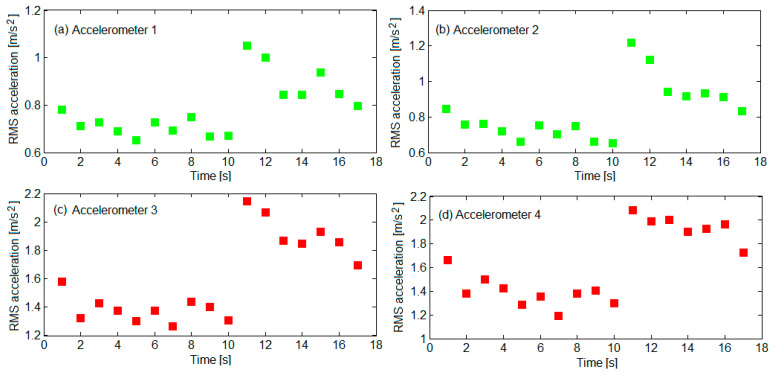
Values of the RMS accelerations for 17 measurement sequences: (**a**) on the bogie frame above the wheelset 1; (**b**) on the bogie frame above the wheelset 2; (**c**) for the axle box of wheelset 1; (**d**) for the axle box of wheelset 2.

**Figure 11 sensors-22-03290-f011:**
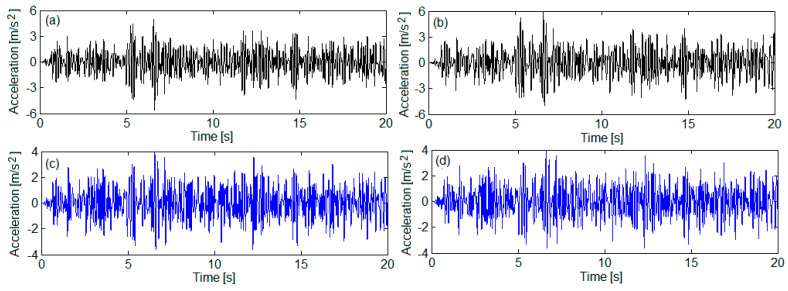
Simulated accelerations for QN1 track quality: (**a**) on the bogie frame against the wheelset 1 suspension—in point *B*_1_; (**b**) on the bogie frame against the wheelset 2 suspension—in point *B*_2_; (**c**) to the axle box of wheelset 1—in point *B_w_*_1_; (**d**) to the axle box of wheelset 2—in point *B_w_*_2_.

**Figure 12 sensors-22-03290-f012:**
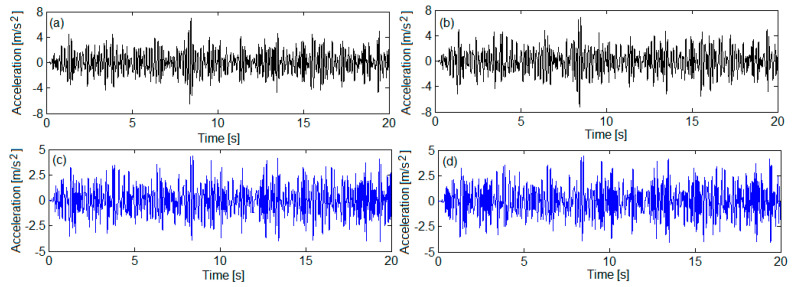
Simulated accelerations for QN2 track quality: (**a**) on the bogie frame against the wheelset 1 suspension—in point *B*_1_; (**b**) on the bogie frame against the wheelset 2 suspension—in point *B*_2_; (**c**) to the axle box of wheelset 1—in point *B_w_*_1_; (**d**) to the axle box of wheelset 2—in point *B_w_*_2_.

**Figure 13 sensors-22-03290-f013:**
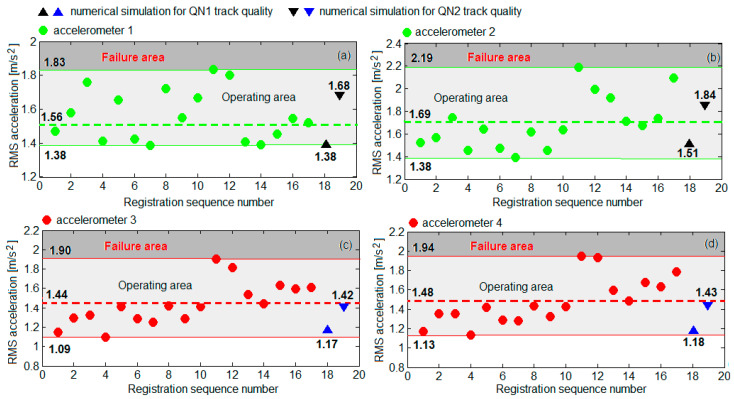
The measured RMS accelerations and the simulated RMS accelerations: (**a**) on the bogie frame against the wheelset 1 suspension; (**b**) on the bogie frame against the wheelset 2 suspension; (**c**) at the axle box of wheelset 1; (**d**) to the axle box of wheelset 2.

**Figure 14 sensors-22-03290-f014:**
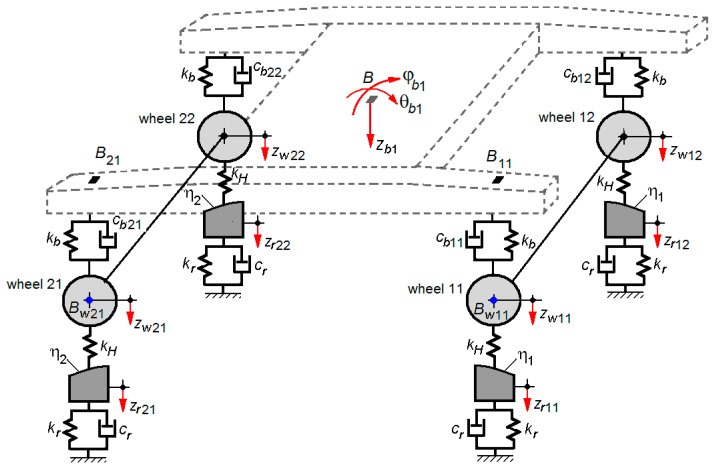
The model of the bogie-track system to study the bogie vertical vibrations at damper failure.

**Figure 15 sensors-22-03290-f015:**
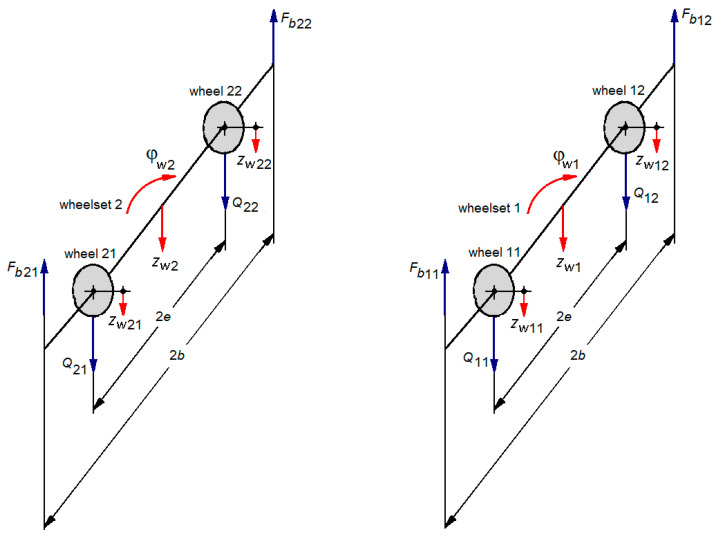
The vibration modes of the bogie wheelsets in Figure 14.

**Figure 16 sensors-22-03290-f016:**
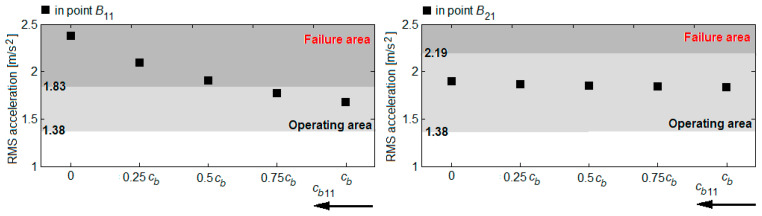
RMS acceleration of the bogie frame upon the damper failure in the wheel 11 suspension.

**Figure 17 sensors-22-03290-f017:**
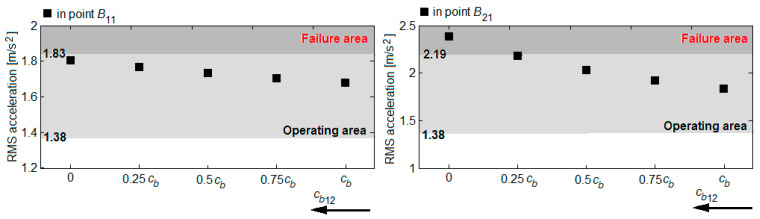
RMS acceleration of the bogie frame upon the damper failure in the wheel 12 suspension.

**Figure 18 sensors-22-03290-f018:**
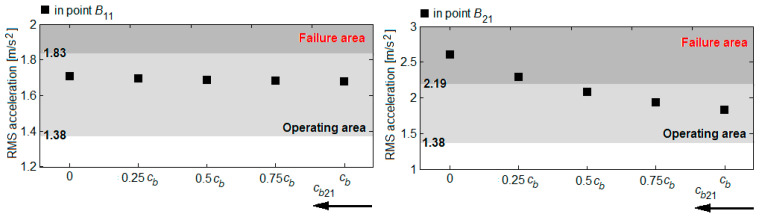
RMS acceleration of the bogie frame upon the damper failure in the wheel 21 suspension.

**Figure 19 sensors-22-03290-f019:**
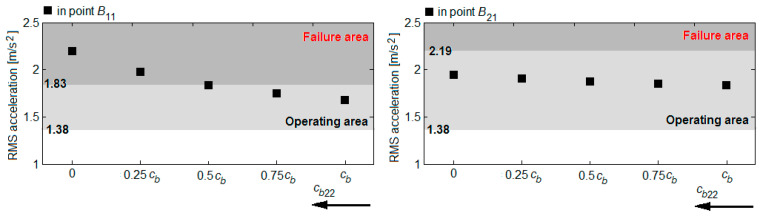
RMS acceleration of the bogie frame upon the damper failure in the wheel 22 suspension.

**Figure 20 sensors-22-03290-f020:**
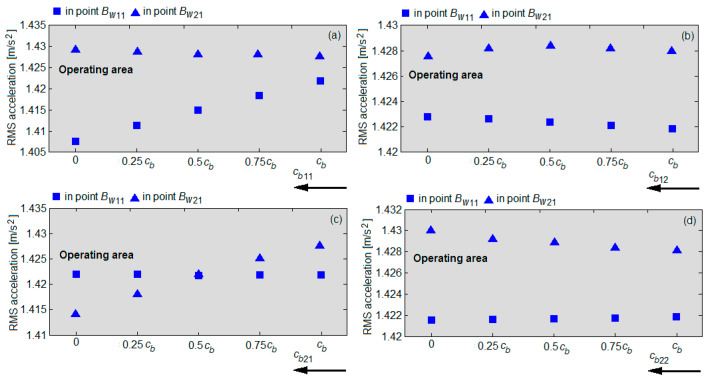
The RMS acceleration of the wheels 11 and 21 upon the damper failure: (**a**) in the suspension of wheel 11; (**b**) in the suspension of wheel 12; (**c**) in the suspension of wheel 21; (**d**) in the suspension of wheel 22.

**Table 1 sensors-22-03290-t001:** Parameters of the numerical model of the vehicle–track system.

Carbody mass	*m_c_* = 41,000 kg
Bogie frame mass	*m_b_* = 2700 kg
Wheelset mass	*m_w_* = 1400 kg
Track mass	*m_r_* = 175 kg
Carbody length	*L_c_* = 24.5 m
Carbody wheelbase	2*a_c_* = 17.2 m
Bogie wheelbase	2*a_b_* = 2.5 m
Carbody pitch inertia moment	*J_c_*_q_ = 1.840 × 10^3^ kg·m^2^
Bogie pitch inertia moment	*J_b_*_q_ = 1.728 × 10^3^ kg·m^2^
Bending modulus	*EI* = 3.158 × 10^9^ Nm^2^
Carbody modal mass	*m_mc_* = 42,477 kg
Carbody modal damping	*c_mc_* = 64.053 kNm/s
Carbody modal stiffness	*k_mc_* = 107.32 MN/m
Elastic constant of the secondary suspension per bogie	2*k_c_* = 1.14 MN/m
Damping constant of the secondary suspension per bogie	2*c_c_* = 81.8 kNs/m
Elastic constant of the primary suspension per wheel	*k_b_* = 0.616 MN/m
Damping constant of the primary suspension per wheel	*c_b_* = 9.05 kNs/m
Elastic constant of the track	*k_r_* = 70 MN/m
Damping constant of the track	*c_r_* = 60 kNs/m
Stiffness of the wheel-rail contact	*k_H_* = 1500 MN/m

**Table 2 sensors-22-03290-t002:** Difference between the simulated RMS acceleration and the average value of the measured RMS accelerations.

Measurement/Simulation Point	for QN1 Track Quality	for QN2 Track Quality
Bogie frame—above wheelset 1	11.54%	7.69%
Bogie frame—above wheelset 2	10.65%	8.87%
Axle box—wheelset 1	18.75%	1.39%
Axle box—wheelset 2	20.27%	3.38%

**Table 3 sensors-22-03290-t003:** The analysis cases of the bogie response to vibrations in case of a damper failure.

Case I (Figure 16)	*c_b_* _11_				*c_b_* _12_	*c_b_* _21_	*c_b_* _22_
	0.75*c_b_*	0.50*c_b_*	0.25*c_b_*	0	*c_b_*	*c_b_*	*c_b_*
Case II (Figure 17)	*c_b_* _12_				*c_b_* _11_	*c_b_* _21_	*c_b_* _22_
	0.75*c_b_*	0.50*c_b_*	0.25*c_b_*	0	*c_b_*	*c_b_*	*c_b_*
Case III (Figure 18)	*c_b_* _21_				*c_b_* _11_	*c_b_* _12_	*c_b_* _22_
	0.75*c_b_*	0.50*c_b_*	0.25*c_b_*	0	*c_b_*	*c_b_*	*c_b_*
Case IV (Figure 19)	*c_b_* _22_				*c_b_* _11_	*c_b_* _12_	*c_b_* _21_
	0.75*c_b_*	0.50*c_b_*	0.25*c_b_*	0	*c_b_*	*c_b_*	*c_b_*

## Data Availability

Not applicable.

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
