# Peer review of "Condition Monitoring of the Dampers in the Railway Vehicle Suspension Based on the Vibrations Response Analysis of the Bogie"

_sensors, 2022, doi:10.3390/s22093290_

Round 1

Reviewer 1 Report

  • The introduction of the paper is too general to give a clear explanation of why the research of the thesis is carried out, and its inherent logic and innovation are not outstanding.
  • It is suggested to simplify the sentences like “but mostly on experience which can lead to maintenance intervals that are much too short.”,
  • Page 2, Line 2, the elements should be changed (or are changed) according to xx
  • Page 4, Line 169, xxx are considered to be equal.
  • The author measured 17 groups of data in practice. Why are the RMS of acceleration greatly different under the same working conditions?
  • In Section 4, why not compare the simulated acceleration with the measured acceleration curve? Please the author to add the time domain curve comparison analysis of acceleration.
  • The author is expected to add the actual measurement results under different conditions, such as two working conditions corresponding to the simulation?
  • Is it reasonable to use only the root mean square value as a criterion to judge the accuracy of the design model? the author is expected to add indicators to illustrate the designed model from many aspects.
  • “The result is a system of 40 first order differential equations”, The author should consider using equation of state in modeling, which can reduce the complexity of formula pushing.

Author Response

Response to the Reviewer 1

First, thank you for taking the time to review the paper and for your comments. The original comments and my responses are listed below. I hope that I have adequately addressed all the comments and that my answers are satisfactory. Modifications and additions were highlighted in the paper by red text.

Comments and Suggestions for Authors 

  • The introduction of the paper is too general to give a clear explanation of why the research of the thesis is carried out, and its inherent logic and innovation are not outstanding.

Reply: I am sorry that the aspects that are underlined at this point of review have not been understood since the introduction of the paper. I will try to remedy this with the explanations listed below:

Regarding the clear explanation of the reason for the research, in the first part of the introduction we presented in detail the need to replace the maintenance of calendar-based railway vehicles with condition-based maintenance, as well as the benefits of condition monitoring. In this context, we framed the problem of detecting and identifying the defects of the suspension system of the railway vehicle and we pointed out the scientific and industrial interest in this current research topic (please see the first seven paragraphs of the Introduction).

Regarding the statement 'and its inherent logic and innovation are not outstanding', I consider it like subjective assessment and unsupported by detailed explanations. The paper includes several contributions (highlighted both in the Introduction and in the Conclusions) which can fit into:

1. Method contributions:

- The principle of the method for detecting the failure of a damper of the primary suspension of the railway vehicle based on the analysis of the bogie response to vibrations, expressed in the terms of RMS acceleration.

- The advantage of the method: the detection of the failure of any of the four dampers of the primary suspension corresponding to a bogie based on the RMS accelerations measured in only two reference points of the bogie chassis.

- The detection of the damper failure for reductions of the damping constant by at least 50%, in the reference point of the bogie chassis located against the faulty damper or in the reference point, diagonally opposite to the faulty damper.

2. Model contributions:

- Development of two original models of the vehicle-track system (rigid-flexible coupled type). These models consider the relevant vertical vibration modes of the car body – bounce, pitch, and the first vertical bending mode, the vertical vibration bounce and pitch of the bogies, the vertical displacements of the wheels and rails and the wheel-track contact elasticity. The first model, with 15 degrees of freedom, is a reference model, used to simulate the bogie’s response to the vertical vibrations for the normal operational condition of the dampers in the primary suspension. The second model, with 20 degrees of freedom, is used to simulate the vibration response of the bogie to the failure of a damper in the primary suspension.

- Validation of the reference model of the vehicle-track system based on the experimental data.

3. New concepts introduced:

- The concept of ‘normal operating area of the damper’ defined by the limits of the interval of dispersion for the RMS accelerations measured in the four reference points of the bogie.

- The concept of ‘failure area of the damper’, delineated inferiorly by the superior limit of the interval of dispersion of the RMS accelerations measured.

4. Original conclusions from the analysis and interpretation of the results.

All these contributions listed above are taken from the paper and can be noticed on a careful reading.

  • It is suggested to simplify the sentences like “but mostly on experience which can lead to maintenance intervals that are much too short.”,

Reply: I corrected. ‘The selection of these intervals is based on theoretical considerations and experience which can lead to too short maintenance intervals.’

  • Page 2, Line 2, the elements should be changed (or are changed) according to xx

Reply: By simplifying this sentence, the clarity of the idea is lost. Correct is 'according to their real condition'  not ‘according to [2]’.

  • Page 4, Line 169, xxx are considered to be equal.

Reply: I modified.

  • The author measured 17 groups of data in practice. Why are the RMS of acceleration greatly different under the same working conditions?

Reply: As I showed in the paper (please see Section 3, under figure 9) 'The dispersion of the RMS accelerations values can be explained via the amplitudes of the track irregularities.' In addition, the conditions are not the same - the track sections are different. I added this mention in introduction of section 3: 'This section describes the architecture of the measurement chain for the bogie vertical accelerations of a passenger vehicle during circulation on a current line at the maximum velocity of 160 km/h, on different track sections in alignment and crosslevel.'

  • In Section 4, why not compare the simulated acceleration with the measured acceleration curve? Please the author to add the time domain curve comparison analysis of acceleration.

Reply: Railway vehicle vibration are random because of track irregularities which are random, and their study requires statistics quantities. Random phenomena are not time repeatable and because of that the comparison of theoretic versus measured time series is not relevant.

  • The author is expected to add the actual measurement results under different conditions, such as two working conditions corresponding to the simulation?

Reply: According with the previous answer, the results for the 17 measurement sequences were compared with the results of the simulations in terms of RMS accelerations (please see figure 12).

  • Is it reasonable to use only the root mean square value as a criterion to judge the accuracy of the design model? the author is expected to add indicators to illustrate the designed model from many aspects.

Reply: Being a real-time monitoring of the suspension damper failure, the root mean square value of the acceleration is a parameter that can be easily measured and compared with a reference value. This parameter was chosen because it is statistically representative. Since the model is linear, the use of other statistical parameters would not provide additional information.

  • “The result is a system of 40 first order differential equations”, The author should consider using equation of state in modeling, which can reduce the complexity of formula pushing.

Reply: Thank you for the suggestion, but I think this form is more appropriate for clarity of exposition.

Reviewer 2 Report

In this paper, the possibility of a new fault detection method for a primary suspension damper of the railway vehicle is investigated. Considering the vibration of car body and bogie with the vertical displacements of the wheel and rails, a simulation model of the vehicle-track system is established and  validated by comparing with the measurement data. On the basis of the above model, the acceleration response of bogie under fault conditions of damper is simulated by changing damping parameter. Simulation results show that failure of damper has a significant impact on the RMS accelerations located on the bogie chassis, which provides a new way for fault detection of damper. This work is generally innovative, but there are still some detail issues need to be elaborated by the author from my opinion.

Below are several suggestions and comments:

  1. In introduction, the model-based approaches and data-driven approaches are introduced. It would be better to give comments on the above-mentioned approaches, which should be related to the work proposed.
  2. In Section 2, author selects the Euler-Bernoulli type for the vehicle model. Comparing to Timoshenko type, only bending moment and lateral displacement are considered in this type, without shear deformation and rotary inertia. Author should illustrate why the Euler-Bernoulli type is selected rather than others and why shear deformation and rotary inertia can be ignored.
  3. In Section 5, the results of measured accelerations on the bogie response to vibrations are discussed exhaustively. The superiority of the proposed method can be illustrated better by the comparison between the proposed method and other methods (like model-based and data-based methods).

Author Response

Response to the Reviewer 2

First, thank you for taking the time to review the paper and for your comments. The original comments and my responses are listed below. I hope that I have adequately addressed all the comments and that my answers are satisfactory. Modifications and additions were highlighted in the paper by red text.

Comments and Suggestions for Authors

In this paper, the possibility of a new fault detection method for a primary suspension damper of the railway vehicle is investigated. Considering the vibration of car body and bogie with the vertical displacements of the wheel and rails, a simulation model of the vehicle-track system is established and validated by comparing with the measurement data. On the basis of the above model, the acceleration response of bogie under fault conditions of damper is simulated by changing damping parameter. Simulation results show that failure of damper has a significant impact on the RMS accelerations located on the bogie chassis, which provides a new way for fault detection of damper. This work is generally innovative, but there are still some detail issues need to be elaborated by the author from my opinion.

Below are several suggestions and comments:

1. In introduction, the model-based approaches and data-driven approaches are introduced. It would be better to give comments on the above-mentioned approaches, which should be related to the work proposed.

Reply: Thank you for the suggestion, but the approaches presented in the introduction have the role of introducing the reader to the broad framework of the issue of detecting and identifying the defects in the suspension system of the railway vehicle.  On the other hand, I would like to mention that the research is in a first stage. The results obtained are encouraging for further research to develop a method for fault detection of a damper in the primary suspension of the railway vehicle, based on the analysis of the vertical vibrations response of the bogie. Given the early stage of the research, the method is not yet finalized and, therefore, a comparison with other methods seems to be premature. In fact, I have shown this throughout the paper:

- please see Abstract: 'This paper investigates the possibility of developing a new method for fault detection of a damper in the primary suspension of the railway vehicle, based on the analysis of the vertical vibrations response of the bogie.'

- please see Introduction: 'The paper investigates the possibility of developing a new method to detect damper faults of the primary suspension of the railway vehicle, which is based on the analysis of the response to the bogie vertical vibrations, expressed as RMS accelerations in four reference points of the bogie.'

- please see Section 5.2: 'The observations above show that there is the premise concerning the possibilities of fault detection in one of the dampers in the vehicle primary suspension, based on the analysis of the bogie response to vibrations, expressed as the RMS acceleration measured only in two reference points of the bogie chassis. '

- please see Conclusions: 'The latest conclusions confirm that it is very likely to develop a fault detection method in a damper of the primary suspension, based on the analysis of the bogie response to vibrations, expressed in the form of RMS acceleration. ' (paragraph added after review).

Last but not least, as you can see, the introduction takes up a considerable amount of space in the paper (two pages), and the required analysis would lead to the excessive expansion of this section.

2. In Section 2, author selects the Euler-Bernoulli type for the vehicle model. Comparing to Timoshenko type, only bending moment and lateral displacement are considered in this type, without shear deformation and rotary inertia. Author should illustrate why the Euler-Bernoulli type is selected rather than others and why shear deformation and rotary inertia can be ignored.

Reply: The "flexible carbody" models of railway vehicles can range from relatively simple Euler-Bernoulli or Timoshenko beams, to structure-type models consisting of plates and beams or complex models obtained on the basis of the multi-body system or finite element method.

This study is dedicated to the bogie vibrations. The influence of the carbody vibrations on the bogie vibrations is manifested by the effect of the rigid vibration mode (bounce) and of the first bending mode in the frequency range of interest (0.5 -20 Hz). Therefore, a complex model of the carbody does not bring new elements in the studied problem.

The Euler-Bernoulli beam model is a model frequently used in the study of the vibrations of the railway vehicle because it leads to numerical results according to those obtained experimentally.

The Timoshenko beam model is useful for representing the higher order flexible modes of the beams when the wavelength of the bending waves is of the same order of magnitude as the radius of inertia of the cross section of the beam. In the case of passenger vehicle, the length of the carbody (modeled as a beam) is much longer than the radius of inertia of the cross section.

3. In Section 5, the results of measured accelerations on the bogie response to vibrations are discussed exhaustively. The superiority of the proposed method can be illustrated better by the comparison between the proposed method and other methods (like model-based and data-based methods).

Reply: The answer to this point is contained in the answer to the first point in the review and please evaluate accordingly. I mentioned above that the research is in a first stage, but the results obtained are encouraging for further research to develop a method for fault detection of a damper in the primary suspension of the railway vehicle, based on the analysis of the vertical vibrations response of the bogie. Given the early stage of the research, the method is not yet finalized and, therefore, a comparison with other methods seems to be premature.

Reviewer 3 Report

In this manuscript, the authors investigated the possibility of developing a new method to detect damper faults of the primary suspension of the railway vehicle, based on the acceleration responses in four reference points of the bogie. The vehicle-track model was established, and the bogie vertical accelerations were measured. Finally the simulated responses were analysed based on the measured results.

The research topic is interesting and within the journal scope. The organization and writing are generally good and clear. It can be considered for publication with considering the following recommendations and suggestions.

  • In the vehicle-track model, some parts of the track system were simplified (e.g. the rails and sleepers). Could you please provide more description on the simplification reason and possible effects on the results?
  • On the wheel-rail coupling relationship, the short-wavelength irregularities (<3 m) and wheel out-of-round wear were not considered. Do they affect the results? Please describe more if possible.
  • Please add the sub-figure number “(a) and (b)” for Figure 4. In addition, it is suggested to provide the PSD of the track irregularities and if possible please compared the PSD presented in this study with some other track irregularity spectra, such as US FTA or Germany hi-speed railway track irregularities.
  • In Table 1, the items “track mass”, “elastic constant of the track” and “damping constant of the track” are not clear. A track includes rail, fastening, sleeper and ballast. What do the word “track” in these items refer to?
  • The conclusion part should be re-organised. Please simplify and list the most important findings of this study.

Author Response

Response to the Reviewer 3

First, thank you for taking the time to review the paper and for your comments. The original comments and my responses are listed below. I hope that I have adequately addressed all the comments and that my answers are satisfactory. Modifications and additions were highlighted in the paper by red text.

Comments and Suggestions for Authors

In this manuscript, the authors investigated the possibility of developing a new method to detect damper faults of the primary suspension of the railway vehicle, based on the acceleration responses in four reference points of the bogie. The vehicle-track model was established, and the bogie vertical accelerations were measured. Finally the simulated responses were analysed based on the measured results.

The research topic is interesting and within the journal scope. The organization and writing are generally good and clear. It can be considered for publication with considering the following recommendations and suggestions.

  • In the vehicle-track model, some parts of the track system were simplified (e.g. the rails and sleepers). Could you please provide more description on the simplification reason and possible effects on the results?

Reply: As I showed, if the coupling effects between the wheels due to the propagation of the bending waves through the rails are neglected, an equivalent model with concentrated parameters is adopted for the track. Frequency range of the railway vehicle is 0.5-20 Hz because the natural frequency of the carbody and bogie on the two-stage suspension are within this range. On the other hand, the resonance frequencies of the track as flexible structure, including flexible rails and sleepers are situate beyond 70-80 Hz. At lower frequencies than the first resonance frequency (70-80 Hz), the bending waves in rails are evanescent and due to that they disappear practically at short distance from the wheel and can't be affect the dynamics of other wheel/rail pairs. Consequently, the track can be modeled using models with concentrated parameters.

  • On the wheel-rail coupling relationship, the short-wavelength irregularities (<3 m) and wheel out-of-round wear were not considered. Do they affect the results? Please describe more if possible.

Reply: The method to synthesize the vertical track irregularities relies on the power spectral density of the track irregularities, as described by ORE B176 [61] and the specifications stipulated by the UIC 518 Leaflet [62] concerning the track geometry quality in the testing and homologation of the railway vehicles from the perspective of the dynamic behaviour. As I shown, according to UIC 518 Leaflet, the standard deviation of the vertical track irregularities which are considered in the method to synthesize the vertical track irregularities correspond to the wavelength ranging from 3 to 25 m. It should be mentioned that the UIC 518 Leaflet is mandatory in the railway vehicle industry. 

For a better understanding of the principle of the method used to synthesize the vertical track irregularities of the path, I added the following explanation at the beginning of the section 2.2:  ‘The method to synthesize the vertical track irregularities [58, 59] relies on the power spectral density of the track irregularities, as described by ORE B176 [61] and the specifications stipulated by the UIC 518 Leaflet [62] concerning the track geometry quality in the testing and homologation of the railway vehicles from the perspective of the dynamic behaviour. According to this method, the vertical track irregularities are described via a pseudo-random function.’

Regarding the fact that wheel out-of-round wear was not taken into account, this does not affect the results. According on the method used, vehicle vibrations induced by track irregularities with wavelengths less than 3 m and greater than 25 m are filtered. On the other hand, the wheel defects have wavelengths of less than 3 m because the wheel circumference is less than 3 m (the wheel diameter is 950 mm) and therefore the wheel defects were neglected.

  • Please add the sub-figure number “(a) and (b)” for Figure 4. In addition, it is suggested to provide the PSD of the track irregularities and if possible please compared the PSD presented in this study with some other track irregularity spectra, such as US FTA or Germany hi-speed railway track irregularities.

Reply: The notations for sub-figures (a) and (b) are located in the upper right corner of Figure 4 (Figure 5, after renumbering).  I added the power spectral density of the vertical track irregularities (please see figure 2) and the following comment: ‘This theoretical curve of the power spectral density is representative for the average statistical properties of the European railway’. Regarding the comparison with other track irregularities spectra, such as US FTA or Germany high-speed railway track irregularities, this question is beyond the paper aim. Moreover, such comparisons have already been presented in the literature (e.g. please see Chapter Track Irregularity Power Spectrum and Numerical Simulation, pp. 137-160, in the book High Speed Railway Track Dynamics, 2017, Publisher Springer).

  • In Table 1, the items “track mass”, “elastic constant of the track” and “damping constant of the track” are not clear. A track includes rail, fastening, sleeper and ballast. What do the word “track” in these items refer to?

Reply: Track dynamic behaviour at level of the rail at low and medium frequency is like two degrees of freedom system with concentrated parameters. Restraining the frequency range to the one of the vehicle, the Single-Degree-of-Freedom System model can be adopted as an equivalent track model (please, see Esveld, C., Modern Railway Track, Delft University of Technology). Parameters of the equivalent model are set so that the model reproduces the stiffness of the track, its natural low frequency and the amplitude resonance.

  • The conclusion part should be re-organised. Please simplify and list the most important findings of this study.

Reply: We simplified the conclusions section by removing sentences that did not contain important findings from this study. For this, I have removed the following paragraph: ‘The numerical simulation applications have been developed on the basis of a rigid-flexible coupled model of the vehicle – track system, with 15 degrees of freedom, which considers the relevant vibration modes in the carbody vertical plan – bounce, pitch and the first vertical bending mode, the bounce and pitch vibrations of the bogie, the vertical displacements of the wheels and rails. To simulate the bogie response to the vertical vibrations upon a damper failure, the vehicle – track system model has been extended to a model with 20 degrees of freedom, where the roll vibrations of the bogie and its wheelsets were taken into account. To validate the theoretical model of the vehicle – track system, the values of the measured RMS accelerations were used as reference, and the percent difference between the RMS accelerations obtained by numerical simulations and the average of the measured RMS accelerations was used as a comparison criterion.’

On the other hand, in order to respond to Reviewer 4, I have completed the conclusions with the following paragraph: ‘The latest conclusions confirm that it is very likely to develop a fault detection method in a damper of the primary suspension, based on the analysis of the bogie response to vibrations, expressed in the form of RMS acceleration.  The strength of this method lies in the fact that it is possible to detect failure in any of the four dampers of the primary suspension, corresponding to a bogie based on the RMS accelerations measured in only two reference points of the bogie chassis.’

Reviewer 4 Report

Dear Author

I like the general approach with numerical and experimental analysis. I am not native speaker, but I recommend some improvements in language.

            The last sentence in Abstract is too long

            Line 75 consider safety instead of security

            Circulation? Line 107

            ….

Most likely typo error in line 196 25 (m)?

Please check the table 1 there are two carbody modal damping with different units (kg?)

Bending modulus unit

And maybe the hardest issue to comment. Did you consider the tolerances of the bogey parts (differences in mass, damping and elastic constants? …) This might have significant influence on failure area. Could at least suggest how these results could be used. Do you plan a test run for the bogey on the test track with full or minimum load?

Best regards

Author Response

Response to the Reviewer 4

First, thank you for taking the time to review the paper and for your comments. The original comments and my responses are listed below. I hope that I have adequately addressed all the comments and that my answers are satisfactory. Modifications and additions are highlighted in the paper by red text.

Comments and Suggestions for Authors

Dear Author,

I like the general approach with numerical and experimental analysis. I am not native speaker, but I recommend some improvements in language.

  1. The last sentence in Abstract is too long.

Reply:  I took into account the observation. I have reduced this sentence: ‘The presented results show that there are clear premises on the possibilities of developing a fault detection method of any of the four dampers of the primary suspension corresponding to a vehicle bogie, based on the RMS accelerations measured only in two reference points of the bogie.’

  1. Line 75 consider safety instead of security.

Reply:  I corrected.

  1. Circulation? Line 107

Reply:  I replaced ‘circulation’ with ‘running’.

  1. Most likely typo error in line 196 25 (m)?

Reply:  I corrected.

  1. Please check the table 1 there are two carbody modal damping with different units (kg?)

Reply:  I corrected. I added ‘Carbody modal mass’.

  1. Bending modulus unit.

Reply:  Unit of measure for bending module is correct. EI=E*I – bending module, where E [N/m2] is the elasticity longitudinal modulus, and I [m4] is the inertia moment of the transversal section.

  1. And maybe the hardest issue to comment. Did you consider the tolerances of the bogey parts (differences in mass, damping and elastic constants? …) This might have significant influence on failure area.

Reply:  I didn't consider the tolerances of the bogie parts (differences in mass, damping and elastic constants …)  because they are very small and do not affect the vibration regime of the bogie.

  1. Could at least suggest how these results could be used. Do you plan a test run for the bogey on the test track with full or minimum load?

Reply:  

As shown in the conclusions, the results of the paper show that damper defects can be detected which lead to the loss of 50% of the damping capacity. Also, I added in Conclusions: 'The latest conclusions confirm that it is very likely to develop a fault detection method in a damper of the primary suspension, based on the analysis of the bogie response to vibrations, expressed in the form of RMS acceleration. The advantage of this method lies in the fact that it is possible to detect failure in any of the four dampers of the primary suspension, corresponding to a bogie based on the RMS accelerations measured in only two reference points of the bogie chassis.'

The experimental results presented in the paper were obtained when running an empty passenger vehicle (minimum load).

Reviewer 5 Report

Manuscript ID: sensors-1688617

In this manuscript, the author investigates the possibility of developing a new method for early damper failure detection mechanism of the primary suspension of the railway vehicle. This technique is basically based on the analysis of the response to the bogie vertical vibrations, expressed as RMS accelerations in four reference points of the bogie. The work is meaningful for display applications, and the paper is well written and organized. However, there are a few comments the author needs to address before it gets published in Sensors.

  • I would recommend the author to add a table that summarizes the introduction and related works and developed models and then compare the present findings to theirs.
  • Most of the sentences are too long. For example “Experimental data are also the basis validating the results of numerical simulations. As for the two models of vehicle track system for simulating the vibrations response of the bogie, both are complex, of rigid-flexible coupled type, which considers the relevant vertical vibration modes of the car body – bounce, pitch, and the first vertical bending, the vertical vibration bounce and pitch of the bogies, the vertical displacements of the wheels and rails and the wheel-track contact elasticity”.
  • It seems that the author neglects the effect of nonlinear stiffness and damping. If yes, could you please add more details describing the reason for neglecting these two parameters?
  • I would advise the author to add a figure just below Equation (8) to show the relative bending eigenmode of the carbody?
  • The second term in Equation (10) is weak. Have tried to scale the EOM and compare the contribution of each term?
  • The experimental and numerical results are clearly presented. I would recommend the author to join them in one graph to show the model accuracy and the error.

Author Response

Response to the Reviewer 5

First, thank you for taking the time to review the paper and for your comments. The original comments and my responses are listed below. I hope that I have adequately addressed all the comments and that my answers are satisfactory. Modifications and additions were highlighted in the paper by red text.

Comments and Suggestions for Authors

In this manuscript, the author investigates the possibility of developing a new method for early damper failure detection mechanism of the primary suspension of the railway vehicle. This technique is basically based on the analysis of the response to the bogie vertical vibrations, expressed as RMS accelerations in four reference points of the bogie. The work is meaningful for display applications, and the paper is well written and organized. However, there are a few comments the author needs to address before it gets published in Sensors.

  • I would recommend the author to add a table that summarizes the introduction and related works and developed models and then compare the present findings to theirs.

Reply: Thank you for the suggestion, but such approach would rather be part of a review paper. On the other hand, I would like to mention, as will be seen in the answer to point 3, that the research is in a first stage. The results obtained are encouraging for further research to develop a method for fault detection of a damper in the primary suspension of the railway vehicle, based on the analysis of the vertical vibrations response of the bogie. Given the early stage of the research, the method is not yet finalized and, therefore, a comparison with other methods seems to be premature. In fact, I have shown this throughout the paper:

- please see Abstract: 'This paper investigates the possibility of developing a new method for fault detection of a damper in the primary suspension of the railway vehicle, based on the analysis of the vertical vibrations response of the bogie.'

- please see Introduction: 'The paper investigates the possibility of developing a new method to detect damper faults of the primary suspension of the railway vehicle, which is based on the analysis of the response to the bogie vertical vibrations, expressed as RMS accelerations in four reference points of the bogie.'

- please see Section 5.2: 'The observations above show that there is the premise concerning the possibilities of fault detection in one of the dampers in the vehicle primary suspension, based on the analysis of the bogie response to vibrations, expressed as the RMS acceleration measured only in two reference points of the bogie chassis. '

- please see Conclusions: 'The latest conclusions confirm that it is very likely to develop a fault detection method in a damper of the primary suspension, based on the analysis of the bogie response to vibrations, expressed in the form of RMS acceleration. ' (paragraph added after review).

I can also add the fact that such tables are already found in other papers (e.g.  ref. 15. Ye, Y.; Huang, P.; Zhang, Y. Deep learning-based fault diagnostic network of high-speed train secondary suspension systems for immunity to track irregularities and wheel wear. Railway Engineering Science 2022, 30, 96–116.)

  • Most of the sentences are too long. For example “Experimental data are also the basis validating the results of numerical simulations. As for the two models of vehicle track system for simulating the vibrations response of the bogie, both are complex, of rigid-flexible coupled type, which considers the relevant vertical vibration modes of the car body – bounce, pitch, and the first vertical bending, the vertical vibration bounce and pitch of the bogies, the vertical displacements of the wheels and rails and the wheel-track contact elasticity”.

Reply: I took into account the observation and modified: “Experimental data are also the basis validating the results of numerical simulations. As for the two models of vehicle track system for simulating the vibrations response of the bogie, both are complex, of rigid-flexible coupled type. These models consider the relevant vertical vibration modes of the car body – bounce, pitch, and the first vertical bending, the vertical vibration bounce and pitch of the bogies, the vertical displacements of the wheels and rails and the wheel-track contact elasticity”.

  • It seems that the author neglects the effect of nonlinear stiffness and damping. If yes, could you please add more details describing the reason for neglecting these two parameters?

Reply: Yes, indeed the effect of the nonlinearity of stiffness and damping was neglected at this point in the research. At this early stage of the research, such simplifying hypotheses can lead to qualitative conclusions that demonstrate from the main point of view the functionality of the method.

  • I would advise the author to add a figure just below Equation (8) to show the relative bending eigenmode of the carbody?

Reply: The shape of the first bending mode of the carbody is well known. This has already been presented in several specialized papers, as well as the excitation mechanism (please see paper Tomioka T., Takigami T., Reduction of bending vibration in railway vehicle carbodies using carbody–bogie dynamic interaction, Vehicle System Dynamics, Vol. 48, Supplement, 2010, pp. 467–486.). In fact, it has already been shown schematically in Figure 1 with a red curved line.

  • The second term in Equation (10) is weak. Have tried to scale the EOM and compare the contribution of each term?

Reply: No, I did not try to scale the EOM and compare the contribution of each term in Equation 10. The general form of the equation of motion of the railway carbody (equation 10), more precisely the left side of the equation, is well known and frequently found in the literature, in studies on the vibrations of railway vehicles (please see the references below). The terms on the right side of the equation change are depending on the secondary suspension model.

    1. Zhou J., Goodall R., Ren L., Zhang H., Influences of car body vertical flexibility on ride quality of passenger railway vehicles, Proceedings of the Institution of Mechanical Engineers, Part F: Journal of Rail and Rapid Transit, Vol. 223, 2009, pp. 461-471.
    2. Gong D., Zhou J.S., Sun W.J., On the resonant vibration of a flexible railway car body and its suppression with a dynamic vibration absorber, Journal of Vibration and Control, Vol. 19, Issue 5, 2013, pp. 649–657.
    3. Gong D., Zhou J., Sun W., Passive control of railway vehicle car body flexural vibration by means of under frame dampers, Journal of Mechanical Science and Technology, Vol. 31, Issue 2, 2017, pp. 555-564.
    4. Jie Chen, Yangjun Wu, Xiaolong He, Limin Zhang, Shijie Dong, Suspension parameter design of underframe equipment considering series stiffness of shock absorber, Advances in Mechanical Engineering 2020, Vol. 12(5), pp. 1–16.

  • The experimental and numerical results are clearly presented. I would recommend the author to join them in one graph to show the model accuracy and the error.

Reply: In Figure 12 (Figure 13 after renumbering) has already been shown the measured RMS accelerations and the simulated RMS accelerations. As for the difference between the simulated RMS acceleration and the average value of the measured RMS accelerations, this can be found in Table 1.

Round 2

Reviewer 1 Report

This paper can now be accepted in its current form.

Reviewer 4 Report

Dear Authors

I have no further comments. I believe your paper is suitable for publishing.

Best regards